# Natural Convection Flow over a Vertical Permeable Circular Cone with Uniform Surface Heat Flux in Temperature-Dependent Viscosity with Three-Fold Solutions within the Boundary Layer

Md Farhad Hasan [1,2], Md. Mamun Molla [3,4,*], Md. Kamrujjaman [5] and Sadia Siddiqa [6]

1   Agriculture Victoria Research, Department of Jobs, Precincts and Regions, Victoria State Government, Bundoora, VIC 3083, Australia; farhad.hasan@agriculture.vic.gov.au
2   School of Computing, Engineering and Mathematical Sciences, La Trobe University, Melbourne, VIC 3083, Australia
3   Department of Mathematics and Physics, North South University, Dhaka 1229, Bangladesh
4   Center for Applied Scientific Computing (CASC), North South University, Dhaka 1229, Bangladesh
5   Department of Mathematics, University of Dhaka, Dhaka 1000, Bangladesh; kamrujjaman@du.ac.bd
6   Artificial Intelligence and Computational Science Laboratory, Pebble AI, Ottawa, ON K2J 6B3, Canada; saadiasiddiqa@gmail.com
*   Correspondence: mamun.molla@northsouth.edu

**Abstract:** The aim of this study is to investigate the effects of temperature-dependent viscosity on the natural convection flow from a vertical permeable circular cone with uniform heat flux. As part of numerical computation, the governing boundary layer equations are transformed into a non-dimensional form. The resulting nonlinear system of partial differential equations is then reduced to local non-similarity equations which are solved computationally by three different solution methodologies, namely, (i) perturbation solution for small transpiration parameter ($\xi$), (ii) asymptotic solution for large $\xi$, and (iii) the implicit finite difference method together with a Keller box scheme for all $\xi$. The numerical results of the velocity and viscosity profiles of the fluid are displayed graphically with heat transfer characteristics. The shearing stress in terms of the local skin-friction coefficient and the rate of heat transfer in terms of the local Nusselt number ($Nu$) are given in tabular form for the viscosity parameter ($\varepsilon$) and the Prandtl number ($Pr$). The viscosity is a linear function of temperature which is valid for small Prandtl numbers ($Pr$). The three-fold solutions were compared as part of the validations with various ranges of $Pr$ numbers. Overall, good agreements were established. The major finding of the research provides a better demonstration of how temperature-dependent viscosity affects the natural convective flow. It was found that increasing $Pr$, $\xi$, and $\varepsilon$ decrease the local skin-friction coefficient, but $\xi$ has more influence on increasing the rate of heat transfer, as the effect of $\varepsilon$ was erratic at small and large $\xi$. Furthermore, at the variable $Pr$, a large $\xi$ increased the local maxima of viscosity at large extents, particularly at low $Pr$, but the effect on temperature distribution was found to be less significant under the same condition. However, at variable $\varepsilon$ and fixed $Pr$, the temperature distribution was observed to be more influenced by $\varepsilon$ at small $\xi$, whereas large $\xi$ dominated this scheme significantly regardless of the variation in $\varepsilon$. The validations through three-fold solutions act as evidence of the accuracy and versatility of the current approach.

**Keywords:** natural convection; computation; temperature-dependent viscosity; uniform heat flux; permeable circular cone; finite difference; boundary layer

## 1. Introduction

Natural convection occurs in the environment. Different industrial applications, closed containers, or any type of chamber, such as a greenhouse, are also good examples of locations where convective heat transfer occurs. In order to characterize different heat transfer applications, heat flux is one of the major indices to be considered. Uniform heat flux is

often considered in geophysics as the soil is a complex porous material. The ability of a soil specimen or soil surface to conduct heat, i.e., thermal conductivity, is significant to agricultural systems, as the growth of plant canopies is directly dependent on soil temperature. Soil has moisture contents, and moisture (water) exhibits temperature-dependent viscosity. The viscosity of water hence directly influences the surface energy balance. Therefore, heat flux through soil or any porous medium is significant in environmental or agricultural applications. Due to different destructive geophysical applications, such as irrigations, compactions, and sampling, the soil often loses its texture. Due to the complexities in the geometry, the soil is often assigned a representative element to discuss different properties, such as electrical or thermal conduction [1–3]. Therefore, heat transfer applications in different geometries representing the soil, or any porous material, have attracted great interest. Researchers have been studying natural convection for decades to solve different heat transfer applications in different industries, such as renewable energy, agriculture, or environment, to name a few [4–9]. In computational fluid dynamics (CFD) research, the study of natural convective flow comprises different geometries and representative elements. The physical models or geometries are often considered to be square or rectangular cavities with heated or cooled walls satisfying certain boundary conditions [10–13]. On the other hand, the representation of elements is particularly defined by different dimensionless parameters which are often varied to conduct sensitivity tests to understand the behavior of the fluid flow under various conditions in order to perform the computation.

The study of heat transfer with different cross-sections has gained strong interest. It is realistic to observe the patterns of fluid flow under various circumstances around surfaces of different cross-sections, such as cylinders or cones, with possible inclusions of permeability [14–18]. Merk and Prins [19,20] are often credited to be the pioneers in developing solutions for the fluid flow past a vertical cone considering an axisymmetric form. Later, Braun et al. [21] and Hering and Grosh [22] studied and developed the numerical model further by keeping the Prandtl number (*Pr*) between 0.7 and 0.72. In any case, all the studies mentioned above agreed on the existence of similar solutions for natural convective flow from the vertical cone. Later, in the 2000s, the series of works done by Hossain and Paul [23,24], as well as Hossain et al. [25], highlighted the investigation of the natural convective flow from a heated vertical permeable circular cone by considering both uniform [25] and non-uniform [23,24] surface heat flux and temperature. The three-fold solutions were obtained by the finite difference method, series solution method, and asymptotic solution method.

Most of the relevant published research works focused on sensitivity analyses on constant viscosity. It is a well-known fact that the viscosity of the fluid does not remain constant as a function of temperature. For example, Cebeci and Bradshaw [26] provided a detailed chart outlining the changes in the water viscosity as a function of temperature, where the value of viscosity decreased by approximately 240% as the temperature of the fluid was elevated from 10 °C to 50 °C. A similar conclusion was drawn for other fluids as well, such as air [26]. Therefore, a proper numerical establishment and computational model in state-of-the-art research should take variable viscosity into account along with other physical attributes. Some of the works have been found in the literature where viscosity was varied, such as [27,28]. Ling and Dybbs [29] mentioned varying viscosity inversely for large *Pr* numbers, but the possible implications of including small *Pr* numbers were not properly investigated. In that case, many important fluids, such as liquid sodium ($Pr \approx 0.004$) and mercury ($Pr \approx 0.03$), are being left out of consideration. Fluids with low *Pr* numbers are thought to have small kinematic viscosity or a greater heat diffusivity. Therefore, fluids with $Pr << 1$ will have heat being diffused quicker than the velocity. As a result, the numerical models that are valid for high *Pr* numbers have significantly less versatility. In terms of variable viscosity with temperature, Rahman et al. [18] investigated the natural convective flow along the vertical wavy cone where viscosity was considered an exponential function of temperature. In a similar geometry, Thohura et al. [30] emphasized the effects of the temperature-dependent thermal conductivity. Recently, Khan et al. [31] studied the entropy

generation of fluids within the incompressible boundary layer to understand the changes in velocity and concentration profiles, which could only be explored due to the consideration of temperature-dependent viscosity. Gladys and Reddy [32] also accentuated the role of the temperature-dependent viscosity of non-Newtonian nanofluids through the accelerating vertical plate to analyze the nonlinear buoyancy impacts.

In the literature, most of the works consider the constant viscosity of the fluid within the boundary layer and failed to prove the accuracy of the model in various ranges of *Pr* numbers. The current study considers both small (for example, 0.05) and large (for example, 0.7) *Pr* numbers. Therefore, it is essential to validate the current approach with different numerical parametric tests to gain more confidence in the accuracy and versatility of the approach. Therefore, more than one numerical solution/validation should be conducted to gain a better understanding of the accuracy. Among different numerical techniques, the Keller box method has been a popular and widely accepted numerical technique for nearly five decades [33], with more applications being added to this scheme recently. For example, Kamran et al. [34] have considered the Keller box approach to describe the Jefferey–Hamel flow by considering different non-dimensional parameters to obtain solutions. Reddy et al. [35] also obtained implicit finite difference results by the Keller box technique to study the Joule heating and associated chemical reactions on the magneto Casson nanofluid. Therefore, another evidence of expanding applications of the Keller box method was duly noted. On the other hand, perturbation and asymptotic solutions were also found to be highly efficient for small and large transpiration parameters ($\xi$), respectively, for the last few years [36]. Therefore, the validations through a combination of conventional and modern approaches should be adequate to prove the accuracy of the current approach and will provide a new dimension to future study.

The present study aims to investigate natural convection flow in a vertical permeable cone with uniform heat surface flux as viscosity varies in the computational model. The present model is also valid for fluids with a low Prandtl number (*Pr*), thus resolving the shortcomings of most of the literature in the past decades. The non-dimensional viscosity-variation parameter ($\varepsilon$) has been included in the model, along with the pseudo-similarity variable ($\eta$), to tune the fluid characteristics in the model. The suction parameter ($\xi$) has been included at various ranges to observe the effect of changes in the rate of heat transfer and shear stress. The sensitivity analyses have been conducted with both constant and variable *Pr* numbers to observe the behaviors of the temperature and velocity distribution of the fluid within the boundary layer. The model has been validated with both fixed and variable *Pr* numbers to showcase the uniformity as well as the accuracy of the approach. The three-fold numerical solutions have been conducted by implementing the aforementioned parameters. The implicit finite difference method together with the Keller box scheme for all $\xi$ has been conducted, followed by perturbation for small $\xi$ and, finally, the asymptotic solution for large $\xi$. To the authors' knowledge, there has not been any published work that considers all the characteristics and scopes mentioned above.

## 2. Formulation of the Problem

A steady two-dimensional free convective flow in laminar form is considered over a vertical permeable circular cone of radius *r* with uniform heat flux. The cone is immersed in a viscous and incompressible fluid with temperature-dependent viscosity, for the numerical investigation. It is assumed that the surface heat flux of the cone is $q_w$. Here, $T_\infty$ is the ambient temperature of the fluid, and the considered geometry is illustrated in Figure 1.

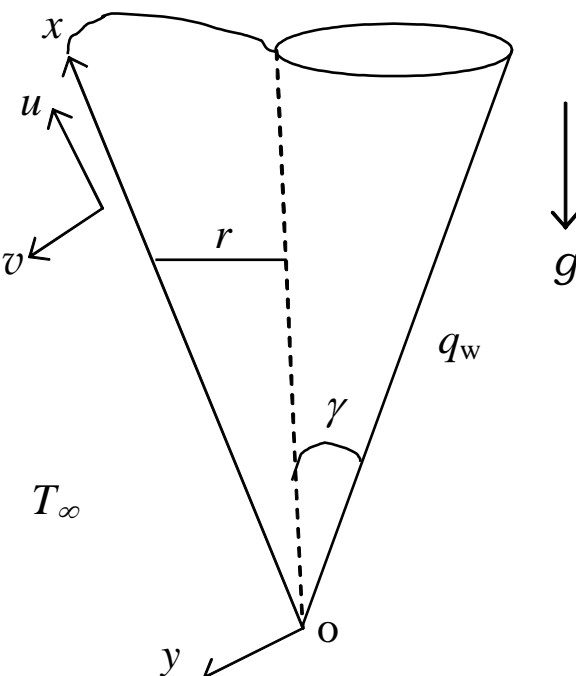

**Figure 1.** Physical geometry and coordinate system of the vertical permeable circular cone considered in this study.

The equations governing the flow are [23–25,30,37]:

$$\text{Continuity equation}: \quad \frac{\partial(ru)}{\partial x} + \frac{\partial(rv)}{\partial y} = 0 \tag{1}$$

$$\text{Momentum equation}: \quad \rho\left(u\frac{\partial u}{\partial x} + v\frac{\partial u}{\partial y}\right) = \frac{\partial}{\partial y}\left(\mu\frac{\partial u}{\partial y}\right) + \rho g\beta(T - T_\infty)\sin\gamma \tag{2}$$

$$\text{Energy equation}: \quad u\frac{\partial T}{\partial x} + v\frac{\partial T}{\partial y} = \frac{k}{\rho C_p}\frac{\partial^2 T}{\partial y^2} \tag{3}$$

The boundary conditions of Equations (1)–(3) are provided below [23]:

$$\left.\begin{array}{l} u = 0, \; v = -V_w, \; q_w = -k\left(\frac{\partial T}{\partial y}\right), \; \text{at } y = 0 \\ u \to 0, \; T \to T_\infty \text{ as } y \to \infty \end{array}\right\} \tag{4}$$

where $(u, v)$ are velocity components along the $(x, y)$ axes, $g$ is the acceleration due to gravity, $\rho$ is the density, $\gamma$ is the cone apex half-angle, $k$ is the thermal conductivity, $\beta$ is the coefficient of thermal expansion, $\mu(T)$ is the temperature-dependent viscosity of the fluid, where $T$ denotes the temperature, and $V$ is the transpiration velocity, which is positive for suction and negative for the injection of fluid through the cone. In the current research, only the suction case has been considered, and therefore $V_w$ has been considered to be positive throughout.

Viscosity variations have multifarious forms. However, the one proposed by Charraudeau [38] is one of the most accepted ones and has been taken into account in this study. The expression could be presented as the following:

$$\mu = \mu_\infty\left[1 + \frac{1}{\mu_{fil}}\left(\frac{\partial\mu}{\partial T}\right)_{fil}(T - T_\infty)\right] \tag{5}$$

where the suffix *fil* represents the film temperature of the fluid.

If suction is non-existent in the model, the boundary layer acts as the free convection boundary layer near the leading edge, although improved downstream suction will be able to dominate the flow to a greater extent. As a result, the following transformations are required [23]:

$$\psi = \nu_\infty r Gr_x^{1/5}\left[f(\xi,\eta) + \tfrac{1}{2}\xi\right], \; T - T\infty = \tfrac{q_w x}{k}Gr_x^{-1/5}\theta(\xi,\eta)$$
$$\eta = \tfrac{y}{x}Gr_x^{1/5}, \; \xi = \tfrac{V_w x}{\nu_\infty}Gr_x^{-1/5}, \; Gr_x = \tfrac{g\beta\cos\gamma x^4}{k\nu_\infty^2}, \; r = x\sin\gamma \tag{6}$$

where $\nu_\infty(=\mu_\infty/\rho)$ is the reference kinematic viscosity, $Gr_x$ is the local Grashof number, $\xi$ is the dimensionless transpiration parameter, $\eta$ is the pseudo-similarity variable, $f(\xi,\eta)$ and $\theta(\xi,\eta)$ are the non-dimensional stream and temperature function, respectively, and $\psi$ is the stream function defined by the following:

$$u = \frac{1}{r}\frac{\partial\psi}{\partial y}, \; v = -\frac{1}{r}\frac{\partial\psi}{\partial x} \tag{7}$$

Substituting (6) into Equations (1)–(5) and after some algebraic calculation, the following transformed equations are obtained:

$$(1+\varepsilon\xi\theta)f''' + \frac{9}{5}ff'' - \frac{3}{5}f'^2 + \varepsilon\xi\theta'f'' + \theta + \xi f'' = \frac{1}{5}\xi\left(f'\frac{\partial f'}{\partial\xi} - f''\frac{\partial f}{\partial\xi}\right) \tag{8}$$

$$\frac{1}{Pr}\theta'' + \frac{9}{5}f\theta' - \frac{1}{5}f'\theta + \xi\theta' = \frac{1}{5}\xi\left(f'\frac{\partial\theta}{\partial\xi} - \theta'\frac{\partial f}{\partial\xi}\right) \tag{9}$$

And the boundary conditions are presented hereby:

$$\left.\begin{array}{l} f = f' = 0, \; \theta' = -1, \text{ at } \eta = 0 \\ f' \to 0, \; \theta \to 0, \text{ as } \eta \to \infty \end{array}\right\} \tag{10}$$

where $\varepsilon$ is the viscosity-variation parameter, and $Pr$ is the Prandtl number defined as the following:

$$\varepsilon = \frac{1}{\mu_{fil}}\left(\frac{\partial\mu}{\partial T}\right)_{fil}\frac{q_w\nu_\infty}{kV} \text{ and } Pr = \frac{\mu_\infty C_p}{k} \tag{11}$$

The local non-similar partial differential equations presented in Equations (8) and (9) need to be solved after finding the boundary conditions (Equation (10)), where the latter could be obtained by the Keller box method, as verified in the literature [25]. The results are computed by considering uniform grids in the $\xi$-direction with 1001 grid-points considered in the $0 \le \xi \le 20$. Further information on iterations and simulation strategies has been included in Section 3.1.

After calculating the respective values of $f$, $\theta$, and their derivatives, the fundamental approach is to calculate the local skin-friction coefficient and the local $Nu$ number from the following expressions:

$$C_{fx}Gr_x^{1/5} = f''(\xi,0) \tag{12}$$

$$Nu_xGr_x^{-1/5} = \frac{1}{\theta(\xi,0)} \tag{13}$$

The results obtained by this method are presented in tabular form in Table 1 for different values of the viscosity variation parameter $\varepsilon$ (=0.0, 0.5, 2.0) and Prandtl number $Pr$ (=0.05, 0.1, 0.7). In the following sections, the solutions for small and large $\xi$ have been presented, and are numerically valid and accurate in both the neighboring leading edge (small $\xi$) and downstream region ($\xi$).

**Table 1.** Comparison of the finite difference solution with asymptotic solutions for the small and large suction parameter $\xi$ in terms of the $C_{fx}Gr_x{}^{1/5}$ and $Nu_x\,Gr_x{}^{-1/5}$ while $Pr = 0.2$ for two different values of $\varepsilon$.

| | $\varepsilon = 0.0$ | | $\varepsilon = 1.0$ | | $\varepsilon = 0.0$ | | $\varepsilon = 1.0$ | |
| | $C_{fx}Gr_x{}^{1/5}$ | | | | $Nu_xGr_x{}^{-1/5}$ | | | |
| $\xi$ | Finite Diff. For All $\xi$ | Small & Large $\xi$ | Finite Diff. For All $\xi$ | Small & Large $\xi$ | Finite Diff. for All $\xi$ | Small & Large $\xi$ | Finite Diff. for All $\xi$ | Small & Large $\xi$ |
|---|---|---|---|---|---|---|---|---|
| 0.0 | 1.9905 | 1.9929 | 1.9905 | 1.9929 | 0.3821 | 0.3821 | 0.3821 | 0.3821 |
| 0.02 | 1.9919 | 1.9958 | 1.9319 | 1.9354 | 0.3838 | 0.3836 | 0.3823 | 0.3822 |
| 0.04 | 1.9939 | 1.9987 | 1.8766 | 1.8823 | 0.3855 | 0.3852 | 0.3826 | 0.3824 |
| 0.06 | 1.9969 | 2.0014 | 1.8256 | 1.8337 | 0.3871 | 0.3867 | 0.3828 | 0.3826 |
| 0.08 | 1.9999 | 2.0042 | 1.7778 | 1.7895 | 0.3887 | 0.3883 | 0.3829 | 0.3827 |
| 0.10 | 2.0028 | 2.0069 | 1.7330 | 1.7499 | 0.3903 | 0.3898 | 0.3833 | 0.3831 |
| 0.14 | 2.0028 | 2.0122 | 1.6909 | 1.6842 | 0.3934 | 0.3929 | 0.3836 | 0.3833 |
| 3.0 | 1.6938 | | 0.3584 | - | 0.6893 | | 0.6591 | - |
| 5.0 | 0.9349 | 0.9388 | 0.1657 | 0.1620 | 1.0185 | 1.0169 | 1.0119 | 1.0095 |
| 6.0 | 0.6741 | 0.6774 | 0.1189 | 0.1145 | 1.2099 | 1.2081 | 1.2077 | 1.2046 |
| 7.0 | 0.5020 | 0.5044 | 0.0898 | 0.0846 | 1.4064 | 1.4044 | 1.4069 | 1.4025 |
| 8.0 | 0.3064 | 0.3284 | 0.0709 | 0.0649 | 1.8044 | 1.6026 | 1.6082 | 1.6014 |
| 10.0 | 0.2488 | 0.2495 | 0.0496 | 0.0416 | 2.0042 | 2.0011 | 2.0163 | 2.0006 |

### 2.1. Solution for Small $\xi$ ($\xi << 1$)

The value $\xi$ for $x$ is small near the leading edge or small $V$ or both, and the series solution of Equations (8) and (9) can be found by applying the perturbation method considering $x$ as a perturbation parameter. In order to establish such an objective, the functions $f(\xi,\eta)$ and $\theta(\xi,\eta)$ in powers of $\xi$ are expanded, and therefore:

$$f(\xi, \eta) = \sum_{i=0}^{\infty} \xi^i f_i(\eta) \text{ and } \theta(\xi, \eta) = \sum_{i=0}^{\infty} \xi^i \theta_i(\eta) \tag{14}$$

Substituting the expansions from Equation (14) into Equations (8) and (9) and equating the various powers of $\xi$ up to $0(\xi^2)$, the following batches of equations are obtained:

$$f_0''' + \frac{9}{5} f_0 f_0'' - \frac{3}{5} f_0'^2 + \theta_0 = 0 \tag{15}$$

$$\frac{1}{Pr} \theta_0'' + \frac{9}{5} f_0 \theta_0' - \frac{1}{5} f_0' \theta_0 = 0 \tag{16}$$

$$f_0(0) = f_0'(0) = 0, \; \theta_0'(0) = -1 \\ f_0'(\infty) = \theta_0(\infty) = 0 \tag{17}$$

$$f_1''' + \varepsilon \big( \theta_0' f_0''' + \theta_0' f_0'' \big) + \frac{9}{5} f_0 f_1'' - \frac{7}{5} f_0' f_1' + 2 f_0'' f_1 + \theta_1 + f_0'' = 0 \tag{18}$$

$$\frac{1}{Pr} \theta_1'' + \frac{9}{5} f_0 \theta_1' - \frac{2}{5} f_0' \theta_1 - \frac{1}{5} \theta_0 f_1' + 2 \theta_0' f_1 + \theta_0' = 0 \tag{19}$$

$$f_1(0) = f_1'(0) = 0, \; \theta_1'(0) = 0 \\ f_1'(\infty) = \theta_1(\infty) = 0 \tag{20}$$

And

$$f_2''' + \varepsilon \big( \theta_0 f_1''' + \theta_1 f_0''' + \theta_0' f_1'' + \theta_1' f_0'' \big) + \frac{9}{5} f_0 f_2'' + \frac{11}{5} f_0'' f_2 \\ - \frac{4}{5} f_1'^2 - \frac{8}{5} f_0' f_2' + 2 f_1'' f_1 + \theta_2 + f_1'' = 0 \tag{21}$$

$$\frac{1}{Pr} \theta_{21}'' + \frac{9}{5} f_0 \theta_2' + \frac{11}{5} \theta' f_{20} - \frac{2}{5} f_1' \theta_1 - \frac{1}{5} \theta_0 f_2' - \frac{3}{5} f_0' \theta_2 + 2 \theta_1' f_1 + \theta_1' = 0 \tag{22}$$

$$f_2(0) = f_2'(0) = 0, \ \theta_2'(0) = 0$$
$$f_2'(\infty) = \theta_2(\infty) = 0 \tag{23}$$

Equations (15)–(23) are solved pair-wise one after another. The solutions are obtained by combining the famous Runge–Kutta–Butcher [39] initial value solver with the Nachts–eim–Swigert iteration scheme [40]. Thus, solutions are found for $f_i$ and $\theta_i$ ($i = 0, 1, 2$) and their respective derivatives.

Once the values of $f_i$ and $\theta_i$ for $i = 0, 1, 2$ and their derivatives are obtained, the local skin-friction coefficient and the local Nusselt number are calculated from the following expressions:

$$C_{fx} Gr_x^{1/5} = f''(\xi, 0) = f_0''(0) + \xi f_1'(0) + \xi^2 f_2''(0) \tag{24}$$

$$Nu_x Gr_x^{-1/5} = \frac{1}{\theta(\xi, 0)} = 1 / \left[ \theta_0(0) + \xi \theta_1(0) + \xi^2 \theta_2(0) \right] \tag{25}$$

The comprehensive values calculated from Equations (24) and (25) are presented in Table 1. The comparison also served as part of the validations of the present approach.

### 2.2. Solution for Large $\xi$

Attention has been given in this part to the behavior of the solution to the Equations (8) and (9) when $\xi$ is considerably large. By the order of magnitude analysis of the various terms, $\xi \theta' f''$ in (8) and $\xi \theta'$ in (9) were found to be the largest. However, in both of their equations, both numerical terms need to be balanced mathematically. The balancing part is performed by assuming $\eta$ to be a small parameter, which will eventually make $\eta$-derivatives larger. It is also important to determine the standard scaling approach considering $\theta = O(\xi - 1)$ as $\xi \to \infty$. After balancing the $f'''$, $\theta$, and $\xi \theta' f''$ terms in (9), $\eta = O(\xi - 1)$ and $f = O(\xi - 4)$ as $\xi \to \infty$ would be found, which would also serve as a confirmation of the accuracy in the balancing. As a result, the following expressions are substituted:

$$f = \xi^{-4} \widetilde{f}(\xi, \widetilde{\eta}), \ \theta = \xi^{-1} \widetilde{\theta}(\xi, \widetilde{\eta}), \ \widetilde{\eta} = \xi \eta \tag{26}$$

Substituting this transformation into Equations (8) and (9), we get the following equations:

$$\left(1 + \varepsilon \widetilde{\theta}\right) \widetilde{f}''' + \xi^{-5} \widetilde{f} \widetilde{f}'' + \widetilde{\theta} + \widetilde{f}'' + \varepsilon \widetilde{\theta}' \widetilde{f}'' = \frac{1}{5} \xi^{-4} \left( \widetilde{f}' \frac{\partial \widetilde{f}'}{\partial \xi} - \widetilde{f}'' \frac{\partial \widetilde{f}}{\partial \xi} \right) \tag{27}$$

$$\frac{1}{Pr} \widetilde{\theta}'' + \xi^{-5} \widetilde{f} \widetilde{\theta}' + \widetilde{\theta}' = \frac{1}{5} \xi^{-4} \left( \widetilde{f}' \frac{\partial \widetilde{\theta}}{\partial \xi} - \widetilde{\theta}' \frac{\partial \widetilde{f}}{\partial \xi} \right) \tag{28}$$

The corresponding boundary conditions are

$$\widetilde{f}(\xi, 0) = \widetilde{f}'(\xi, 0) = 0, \ \widetilde{\theta}'(\xi, 0) = -1$$
$$\widetilde{f}'(\xi, \infty) = 0, \ \widetilde{\theta}(\xi, \infty) = 0 \tag{29}$$

where primes describe differentiation with respect to $\widetilde{\eta}$. Equations (27) and (28) are solved in terms of an inverse power of series in $\xi$.

The functions $\widetilde{f}(\xi, \widetilde{\eta})$ and $\widetilde{\theta}(\xi, \widetilde{\eta})$ are expanded in the power series in the negative powers of $\xi$, considering that $\xi$ is large, which yields the following:

$$\widetilde{f}(\xi, \widetilde{\eta}) = \sum_{i=0}^{\infty} \xi^{-5i} \widetilde{f}_i(\widetilde{\eta}) \ \text{and} \ \widetilde{\theta}(\xi, \widetilde{\eta}) = \sum_{i=0}^{\infty} \xi^{-5i} \widetilde{\theta}_i(\widetilde{\eta}) \tag{30}$$

Now substituting the above expansions into Equations (27)–(29) and equating the coefficient of power of $\xi^0$ and $\xi^1$, we get the following equations

$$\left(1 + \varepsilon \widetilde{\theta}_0\right) \widetilde{f}_0''' + \widetilde{f}_0'' + \widetilde{\theta}_0 + \varepsilon \widetilde{\theta}_0' \widetilde{f}_0'' = 0 \tag{31}$$

$$\frac{1}{Pr}\widetilde{\theta}_0'' + \widetilde{\theta}_0' = 0 \tag{32}$$

$$\widetilde{f}_0(0) = \widetilde{f}_0'(0) = 0, \ \widetilde{\theta}_0'(0) = -1$$
$$\widetilde{f}_0'(\infty) = 0, \ \widetilde{\theta}_0(\infty) = 0 \tag{33}$$

$$\left(1 + \varepsilon\widetilde{\theta}_0\right)\widetilde{f}_1''' + \widetilde{f}_0\widetilde{f}_0'' + \widetilde{\theta}_1 + \varepsilon\left(\widetilde{\theta}_1\widetilde{f}_0'' + \widetilde{\theta}_1'\widetilde{f}_0'' + \widetilde{\theta}_0'\widetilde{f}_1''\right) = 0 \tag{34}$$

$$\frac{1}{Pr}\widetilde{\theta}_1'' + 2\widetilde{\theta}_0'\widetilde{\theta}_1' + \widetilde{f}_0'\widetilde{\theta}_0' + \widetilde{\theta}_1' = 0 \tag{35}$$

$$\widetilde{f}_1(0) = \widetilde{f}_1'(0) = 0, \ \widetilde{\theta}_1'(0) = 0$$
$$\widetilde{f}_1'(\infty) = 0, \ \widetilde{\theta}_1(\infty) = 0 \tag{36}$$

The local skin-friction and the local Nusselt number are as follows:

$$C_{fx}Gr_x^{1/5} = f''(\xi,0) = \xi^{-2}\left[\widetilde{f}_0''(0) + \xi^{-5}\widetilde{f}_1''(0)\right] \tag{37}$$

$$Nu_xGr_x^{-1/5} = \frac{1}{\theta(\xi,0)} = \frac{\xi}{\widetilde{\theta}_0(0) + \xi^{-5}\widetilde{\theta}_1(0)} \tag{38}$$

The asymptotic solutions obtained from (31)–(36) in terms of local skin-friction and local Nusselt number and compared with the solution of the finite difference method in Table 1.

## 3. Results and Discussion

### 3.1. Computational Methods

The analyses have been conducted by tuning the non-dimensional empirical parameters at different stages with both fixed and variable $Pr$ numbers. The computation was done by Fortran 90 [41]. The finite difference solutions were obtained by the Keller box scheme [31–42], which is considered to be one of the most accurate implicit finite difference techniques in computational mathematics. Refer to [37] for a detailed explanation of the Keller box method. In this method, the non-linear system of the partial differential equations governing the fluid flow is solved, and an assumption is made in terms of the functions and the derivatives to express the first-order equations. The derivatives in terms of the central differences are approximated in both co-ordinate directions. This part is particularly obtained by denoting the mesh points in the $(\xi, \eta)$ plane by $x_i$ and $\eta_I$, where $i = 1, 2, 3, \ldots, M$ and $j = 1, 2, 3, \ldots, N$ [37], followed by central difference approximations where the equations containing x are centred explicitly at $(\xi_{i-1/2}, \eta_{j-1/2})$ and the rest at $(\xi_i, \eta_{j-1/2})$, where $\eta_{j-1/2} = (\eta_j + \eta_{j-1})/2$, for instance. It yields a batch of non-linear differential equations for the unknowns at xi in as a function of $\xi_{i-1}$. Then, those equations are subject to linearization by Newton's quasi-linearization method with a view to solving them by taking a block-tridiagonal algorithm into account. The initial iteration of the converged solution is considered to be at $\xi = \xi_{i-1}$. At the commencement, i.e., $\xi = 0$, the guess profiles for all the considered variables are provided, and then the Keller box method comes into the computation to solve the governing ordinary differential equations. By obtaining the lower stagnation point solution, the steps could be performed along with the boundary layer of the geometry [37]. The iteration is terminated once the target difference in velocity and temperature computation is reached, and in this study it was $|\delta f^i| \leq 10^{-6}$, where $i$ represents the number of iterations. It should be mentioned here that computations were not conducted by a uniform grid in the $y$-direction. However, a non-uniform grid was considered, which was then defined by $\eta_j = \sinh((j-1)/p)$, with $j = 1, 2, \ldots, 301$ and $p = 100$.

### 3.2. Overview of Numerical Analyses

At first, the validations were conducted by comparing the three different types of solutions considered in this study. A different type of 3D analyses was conducted to assess the

correlation coefficient, and good $R^2$ values were obtained overall. As part of the discussion, the streamlines and isothermal behavior have been presented. After that, the influence of parameters has been investigated on the viscosity, velocity, and temperature distributions. At the end of each analysis, a conclusion has been drawn outlining the significant and non-significant impact of specified parameters under different circumstances.

*3.3. Comparing Finite Difference Solutions with Perturbation and Asymptotic Solutions*

3.3.1. Validation and Comparison at Fixed *Pr*

To assess the accuracy of the approach, the typical finite difference solutions have been compared against the present approach with small and large suction parameters ($\xi$). To achieve such an objective, two different values of $\varepsilon$ were assigned (0 and 2), and *Pr* = 0.7 was considered. In other words, the comparisons have been made in both the absence and presence of $\varepsilon$ for the same type of fluid. The cross-validations were conducted by observing the values of two different parameters namely, the local skin-friction coefficient ($C_{fx}Gr_x^{1/5}$) and the local Nusselt number ($Nu_xGr_x^{-1/5}$). Table 1 contains the comprehensive comparisons of the aforementioned conditions. In general, there is hardly any big difference between the finite difference solutions and the asymptotic solutions. The slight deviation in the values could be attributed to the percentage error in the approach.

According to Table 1, as $\xi$ increased, $C_{fx}Gr_x^{1/5}$ decreased, regardless of the $\varepsilon$ values. However, $C_{fx}Gr_x^{1/5}$ values were found to be lower in the presence of $\varepsilon$, thus confirming the role of the viscosity-variation parameter on the reduction of the local skin-friction coefficient. This behavior could be explained in terms of the viscosity and temperature of the fluid. As $\varepsilon$ increased, the fluid temperature within the boundary layer increased, which led to increased viscosity. Therefore, $C_{fx}Gr_x^{1/5}$ decreased further. Similar patterns were observed for both finite difference and asymptotic solutions.

Meanwhile, $Nu_xGr_x^{-1/5}$ exhibited erratic trends. In general, $Nu_xGr_x^{-1/5}$ increased as $\xi$ increased when $\varepsilon$ = 0 was considered. However, as $\varepsilon$ = 2 was assigned, $Nu_xGr_x^{-1/5}$ values had a different pattern. Initially, $Nu_xGr_x^{-1/5}$ kept on increasing as $\xi$ increased when $\varepsilon$ = 2, and the values of local *Nu* number were lower than those achieved at $\varepsilon$ = 0. However, at $\xi \geq 2$, the values of $Nu_xGr_x^{-1/5}$ exhibited values higher than $Nu_xGr_x^{-1/5}$ at $\varepsilon$ = 0.

Furthermore, surface analyses were conducted to understand the rational behavior as well as the accuracy of the approach. The purpose was to add one type of extra validation in the current approach to check the model predictions for a different type of fluid. Therefore, the local skin-friction coefficients and local *Nu* number were calculated at *Pr* = 0.7 and the numerical accuracy was assessed by observing the coefficient of correlation ($R^2$) at $\varepsilon$ = 0 and $\varepsilon$ = 2. The surface analyses were performed by the data analysis software Origin Pro, developed by OriginLab Corporation [42]. In the 3D analyses, solutions from the finite difference method and perturbation-asymptotic solutions were illustrated in the same frame to observe the analytical behavior as a function of the suction parameter ($\xi$). In general, an $R^2$ between 0.97 and 0.98 was obtained, which provides more confidence in the calculative approach. Figure 2 depicts the predicted surfaces for $C_{fx}Gr_x^{1/5}$ at different $\xi$ in the absence of $\varepsilon$ (Figure 2a) and with an assigned value of $\varepsilon$ (Figure 2b). Most of the obtained points were found on the surface. In short, the marginal calculation error was noticed. On the other hand, a similar analysis was conducted in terms of $Nu_xGr_x^{-1/5}$ in Figure 3 under the same $\varepsilon$ conditions. It is evident that $Nu_xGr_x^{-1/5}$ is increasing concurrently as $\xi$ was increasing in Figure 3a,b, in line with the argument presented in Table 1, where *Pr* = 0.2. Furthermore, the calculated values from the asymptotic and perturbation solutions for small and large $\xi$ were found to be closer than those from the finite difference approach. The missing values have not been included in the illustrations, and therefore the input sample size is not consistent in all figures.

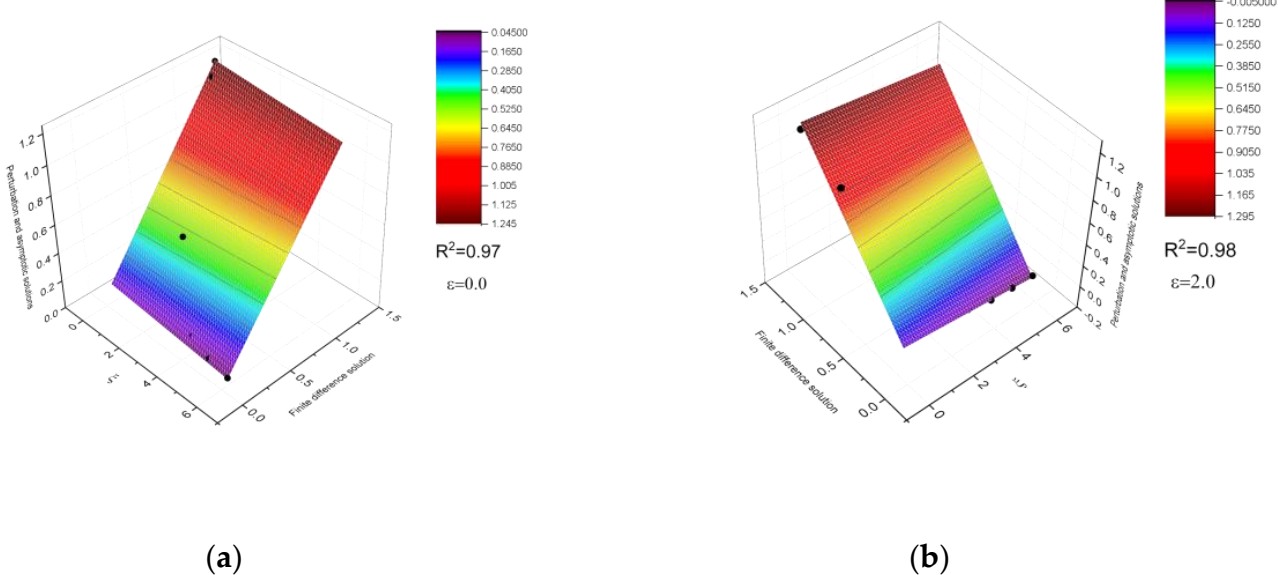

(**a**)                                                        (**b**)

**Figure 2.** Comparison of the finite difference solution with asymptotic solutions for the small and large suction parameter $\xi$ in terms of the $C_{fx}Gr_x^{1/5}$ while $Pr = 0.7$ for two different values of $\varepsilon$, (**a**) $\varepsilon = 0$, and (**b**) $\varepsilon = 2.0$.

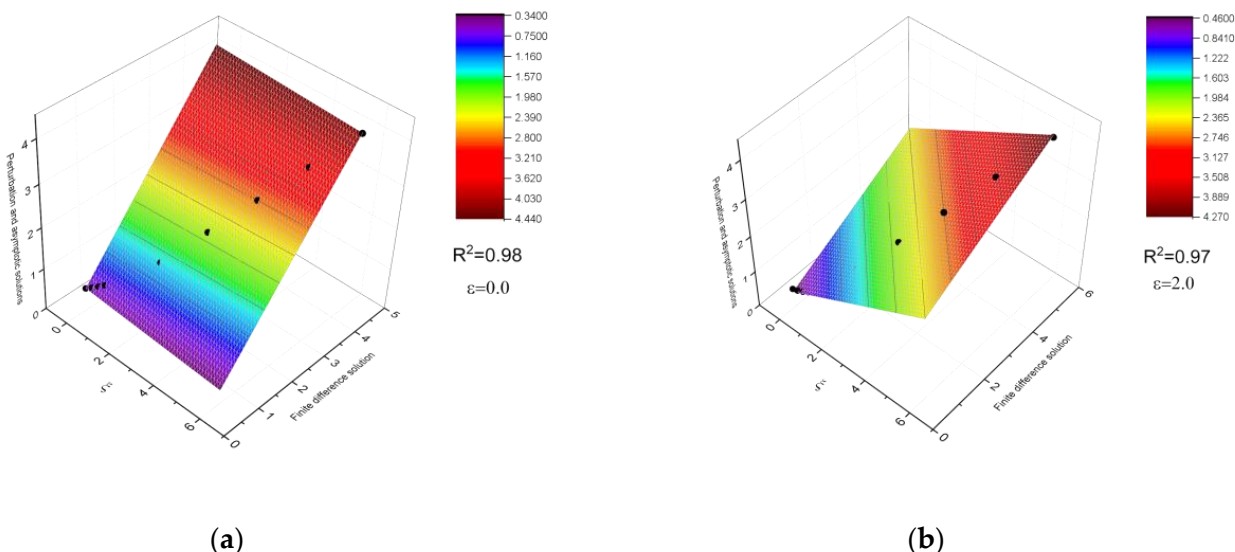

(**a**)                                                        (**b**)

**Figure 3.** Comparison of the finite difference solution with asymptotic solutions for the small and large suction parameter $\xi$ in terms of the $Nu_xGr_x^{-1/5}$ while $Pr = 0.7$ for two different values of $\varepsilon$ namely, (**a**) $\varepsilon = 0.0$, and (**b**) $\varepsilon = 2.0$.

### 3.3.2. Comparison at Variable *Pr*

Figure 4 depicts the comparison of solutions at two different *Pr* values to confirm the accuracy of the approach for two different types of fluids. It could be observed that the agreements were excellent in terms of both $C_{fx}Gr_x^{1/5}$ (Figure 4a) and $Nu_xGr_x^{-1/5}$ at $\xi \in [0,20]$ (Figure 4b)

It was observed that as *Pr* increased $C_{fx}Gr_x^{1/5}$ values decreased, while $Nu_xGr_x^{-1/5}$ increased concurrently. As $\xi$ increased, the temperature of the fluid increased as mentioned earlier, and hence at $\xi = 20$ the values of $C_{fx}Gr_x^{1/5}$ were close to 0 due to the dominance of the suction parameter, whereas at the same suction parameter value, $Nu_xGr_x^{-1/5}$ was found to be the highest regardless of *Pr* values. However, it should be mentioned that as

$\xi$ kept increasing, the difference between the $Nu_x Gr_x^{-1/5}$ of the fluid corresponding to $Pr = 0.1$ became wider than the former, with $Pr = 0.05$.

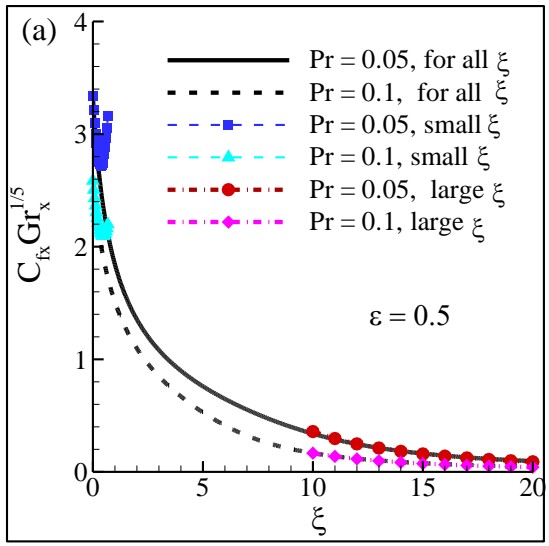 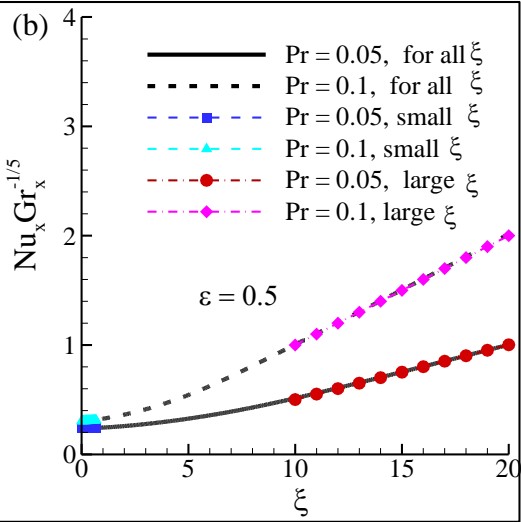

**Figure 4.** Comparison of the finite difference solution with asymptotic solutions for small and large suction parameters $\xi$ in terms of (**a**) the local skin-friction coefficient ($C_{fx} Gr_x^{1/5}$) and (**b**) the local Nusselt number ($Nu_x Gr_x^{-1/5}$) while $\varepsilon = 0.5$ and for two different values of $Pr$.

Since the temperature of the fluid kept increasing at increasing $\xi$ values, the shear stress, i.e., the local skin-friction coefficient, decreased, and to balance the temperature difference, the rate of the heat transfer, i.e., the local $Nu$ number, increased.

### 3.4. Development of Streamlines

This part of the discussion refers to fluids corresponding to two different types of $Pr$ numbers ($Pr = 0.05, 0.1$) to depict the changes in the streamlines as a function of the suction parameter $\xi$. The streamlines illustrate the flow pattern within the momentum boundary layer. The changes were analyzed by observing the variations in the values of the flow rate ($\psi$). Two different viscosity-variation parameters ($\varepsilon = 0.0, 0.5$) were considered for each type of fluid to include the effect of $\varepsilon$ on the streamlines. It was anticipated that the differences in the shapes of streamlines in both $\varepsilon$ values would be quite marginal at low $\xi(\xi < 4)$, due to the lower effect of the suction parameter. However, as $\xi$ increases to a considerably larger value ($\xi > 4$), the differences between the shapes of streamlines of $\varepsilon = 0.0$ and $\varepsilon = 0.5$ increase and do not overlap. Furthermore, since $\varepsilon$ is the viscosity-variation parameter, its absence will lead to the maximum flow rate, regardless of the $Pr$ values. This behavior was expected due to the effect of viscosity on the fluid movement, which has been explained in the later sections, where the distribution of velocity profiles has been comprehensively discussed.

According to Figure 5, at lower $\xi$ the streamlines corresponding to both $\varepsilon$ values overlap, indicating the non-significant impact of the suction parameter on the fluid characteristics. A similar behavior was found for both $Pr = 0.05$ (Figure 5a) and $Pr = 0.1$ (Figure 5b). However, as $\xi$ kept on increasing, the differences in the maximum values of the fluid flow rate of each segment corresponding to $\varepsilon = 0.0$ and $\varepsilon = 0.5$ were increasing gradually, which was mentioned in the first paragraph. This behavior could be also explained in terms of the effect of $\varepsilon$ on the streamline's development. In the presence of the viscosity-variation parameter, the streamlines of the fluid seem to occupy less surface area due to the effect of $\varepsilon$, which eventually restricts the local maximum of the velocity distribution of the fluid. As per Figure 5a, while the maximum flow rate was found to be 42.90 in the absence of $\varepsilon$, the value plummeted to 13.11 (Figure 5b). On the other hand, the fluid corresponding to $Pr = 0.1$ exhibited a different trend in the pattern of streamlines. While the streamlines

corresponding to the maximum flow rate were more aligned to the right side of the boundary layer, the maximum flow rate curve at $Pr = 0.1$ shifted to the center of the boundary, indicating the effect of $Pr$ on the fluid characteristic. If $Pr$ is increased, the momentum and thickness of the thermal boundary layer decreased, and hence the flow rate got reduced.

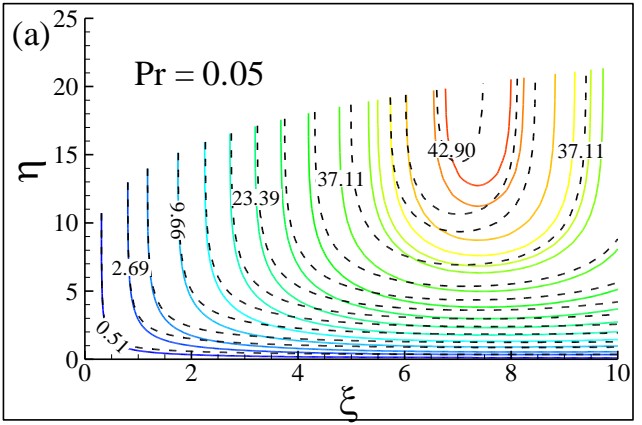 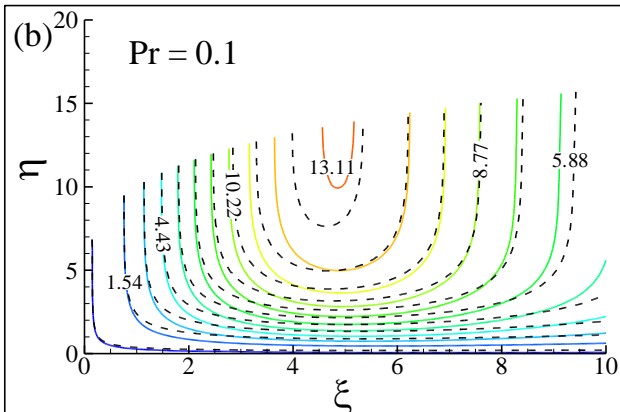

**Figure 5.** Streamlines for the two different viscosity-variation parameters $\varepsilon$ at (**a**) $Pr = 0.05$ (**b**) $Pr = 0.1$. Here, the solid lines stand for $\varepsilon = 0.0$ and the dashed lines for $\varepsilon = 0.5$. The values on each curve represent the flow rate ($\psi$).

*3.5. Isothermal Behavior within the Boundary Layer*

After investigating the behavior of the streamlines, a similar parametric effect was studied in terms of isotherms, illustrated in Figure 6. The isotherms depict the temperature distribution within the boundary layer. It was expected that the isothermal lines closer to the wall would be equal to the boundary temperature, which would lead to lines without any significant peak values. The isothermal line considerably far from the wall boundary would have peak values before plummeting gradually as the lines tended to come near the boundary layer.

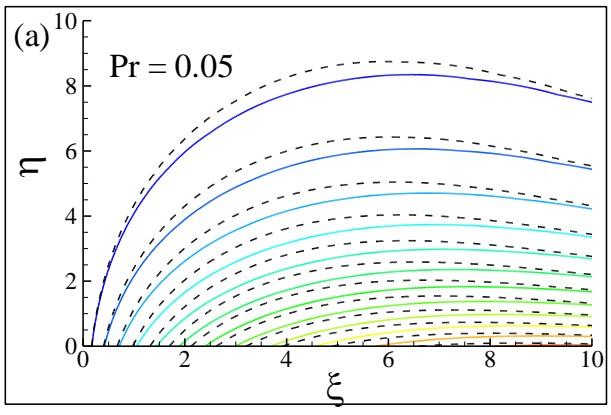 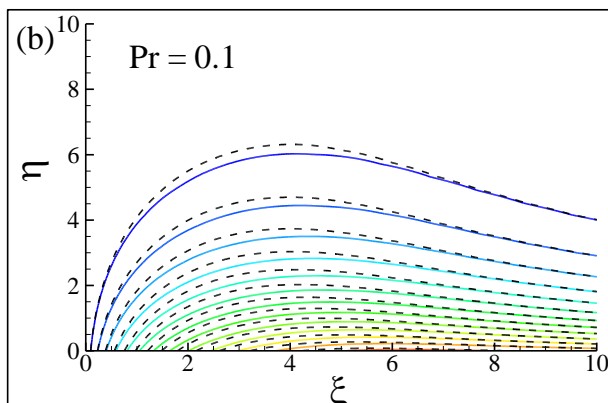

**Figure 6.** Isotherms for the two different viscosity-variation parameters $\varepsilon$ at (**a**) $Pr = 0.05$ (**b**) $Pr = 0.1$. Here, the solid lines stand for $\varepsilon = 0.0$ and the dashed lines for $\varepsilon = 0.5$.

The isotherms appended in Figure 6 confirm the aforementioned statement. The isothermal lines far from the boundary layer could be seen with a peak before gradually starting to reduce. There is a slight deviation in the values between $\varepsilon = 0.0$ and $\varepsilon = 0.5$, where the presence of $\varepsilon$ reasonably increased the isothermal lines due to the influence of the viscosity-variation. However, this was only visible when the lines were at an optimal distance from the thermal boundary layer, where the influence of the latter would be less

significant. A similar behavior was obtained in terms of streamlines as well where the presence of $\varepsilon$ shifted the streamlines up compared to the streamlines corresponding to $\varepsilon = 0.0$. However, as the fluid motion decreases with an increase from $Pr = 0.05$ (Figure 6a) to $Pr = 0.1$ (Figure 6b), the peak values of isothermal lines decrease (visible ones) due to the reduction of the flow rate. The significant reduction in the maximum flow rate was also recorded in Figure 6b, and hence the consistency of the streamlines and isothermal analyses could be confirmed before heading towards the detailed parametric analyses in the following sections.

### 3.6. Impact of $\varepsilon$, $\eta$, and $\xi$ Fluid Characteristics at Fixed Pr

3.6.1. Influence on Viscosity Distribution

Figure 7 depicts the effects of both the viscosity-variation parameter ($\varepsilon$) and the suction parameter ($\xi$) on the dimensionless viscosity. The pseudo-similarity variable ($\eta$) was varied from 0 to a minimum of 6 (for $\xi = 10$) and a maximum of 10.0 ($\xi = 1$), depending on where the viscosity started to remain indifferent regardless of the $\eta$ values. In Figure 7a,b, $Pr = 0.1$ was considered to maintain the consistency of the fluid characteristics.

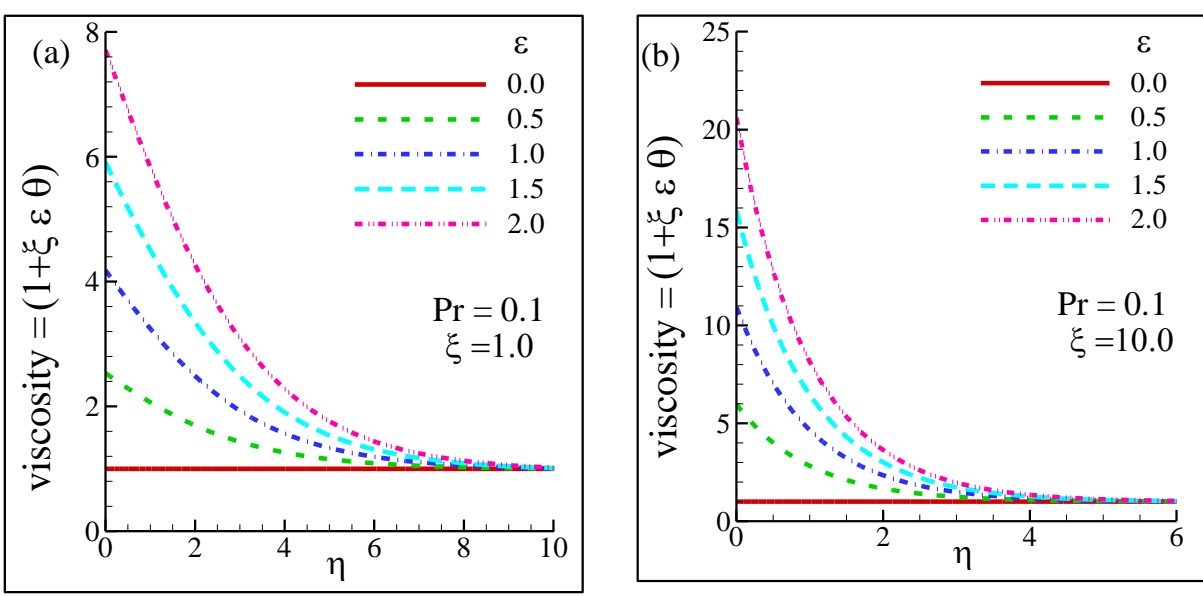

**Figure 7.** Viscosity distribution profiles for different viscosity-variation parameters $\varepsilon$ at (**a**) suction parameter $\xi = 1$ and (**b**) suction parameter $\xi = 10$ while $Pr = 0.1$.

In general, it was found that as $\eta$ increased, viscosity exhibited the local maximum values at the maximum $\varepsilon$, in this case $\varepsilon = 2$, as well as at $\eta = 0$. However, the local maximum viscosity was obtained at approximately 20.5 at $\xi = 10$, whereas it was found to be around 7.6 at $\xi = 1$. This behavior could be attributed to the fact that as the suction parameter increases, the viscosity exhibits the maximum values due to the absence of a pseudo-similarity variable which acts as a counterincentive to viscosity. On the other hand, $\xi$ works as a catalyst to the viscosity at an increased suction parameter, and at $\eta = 0$, the temperature of the fluid within the respective boundary layer decreases, leading to the increased value of viscosity. Meanwhile, the curves corresponding to $\varepsilon \in [0,2]$ in Figure 7a,b showed similar decreasing trends in viscosity. At $\varepsilon = 0$, there is no variation in the viscosity, and therefore $\eta$ will always be 1, regardless of the values of other influential parameters. As $\varepsilon$ began to increase, peak values started to increase from $\eta = 0$, before merging towards the unity, indicating an insignificant impact of $\eta$ on the viscosity. Due to the dominance of $\xi = 10$, the viscosity reached the unity at a lower $\eta$ than that of $\xi = 1$, which was also anticipated.

### 3.6.2. Effect on Velocity Distribution ($f'$)

In this part of the study, the changes in the non-dimensional velocity profiles were investigated by varying $\eta$ and $\varepsilon \in [0,2]$, at $Pr = 0.1$.

It was anticipated that, in the absence of $\varepsilon$, the velocity distribution would keep increasing as $\eta$ increased and quickly reached the peak value, before $\eta$ started to dominate, which would lead to a rapid decrease in the velocity ($f'$). As a consequence, at one specific value of $\eta$, the fluid would not exhibit any rapid mobility within the boundary. However, the local maximum of velocity would occur at a later stage of $\eta$ if $\varepsilon$ kept increasing. Furthermore, the static condition of fluid was also expected to be quicker for a higher $\xi$ (in this case, $\xi = 10$), as a higher suction parameter demobilizes the fluid motion to a greater extent. Figure 8 illustrates the influence of the aforementioned parameters on the velocity distributions. As per Figure 8, as $\varepsilon$ keeps increasing, the curves corresponding to the velocity exhibited peak values and later started to decrease significantly, heading to the static state as $\eta$ increased.

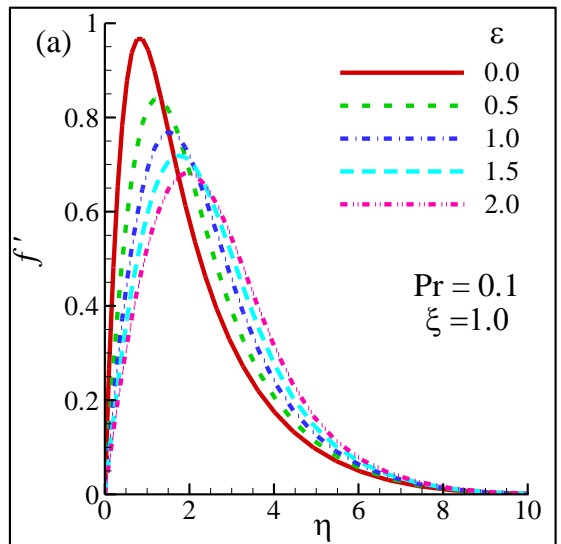 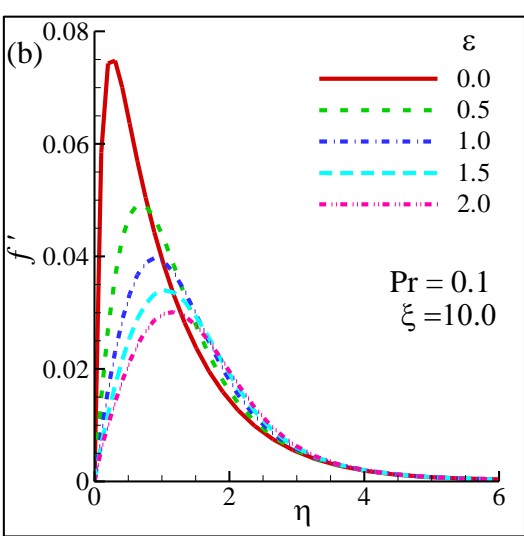

**Figure 8.** Changes in velocity distribution profiles for different viscosity-variation parameters $\varepsilon$ at (**a**) suction parameter $\xi = 1$ and (**b**) suction parameter $\xi = 10$ while $Pr = 0.1$.

At $\varepsilon = 0$ and $\xi = 1$ (Figure 8a), the velocity reached its maximum value of 0.955 at $\eta = 1$, whereas it was 0.85 at $\varepsilon = 0.5$. The lowest peak was recorded at the maximum $\varepsilon$ values considered in this section ($\varepsilon = 2$), and due to the dominance of $\eta$ and $\varepsilon$, the fluid had the lowest peak. In any case, after reaching the peak, velocity started to decrease significantly before reaching the completely static state at $\eta = 10$. On the other hand, at $\xi = 10$ (Figure 8b), similar trends were observed for the velocity at different $\varepsilon$ and $\eta$, but due to the 10-time-increasing high suction parametric value, the peak velocity values were significantly lower, which indicates the superiority of the high $\xi$ values. It is also expected that further increases in $\xi$ will lead to further reduced peak values, and, at one stage, the fluid will remain static.

### 3.6.3. Changes in Temperature Distribution

Studying the changes in the temperature was crucial to the current research, since it provided more information on the effect of $\varepsilon$ and $\eta$. To maintain consistency in the presentation, the changes in the non-dimensional temperature were studied by varying $\eta$ and $\varepsilon \in [0,2]$, at $Pr = 0.1$.

Figure 9 reports the consistent decreasing trend of temperature as $\eta$ increased. However, the peak $\theta$ was recorded at the highest $\varepsilon$ (in this case $\varepsilon = 2$) and lowest pseudo-similarity variable ($\eta = 0$), where $\xi$ was unity. This behavior could be well-explained in the light

of a conceptual understanding of the effect of the viscosity-variation parameter on the temperature. In the absence of $\eta$ and $\varepsilon$, the temperature exhibited the lowest peak value of 2.9, whereas at $\varepsilon = 2$ the value of dimensionless temperature was found to be 3.35. The differences in the $\theta$ outline the impact of the viscosity-variation parameter on the temperature. However, as $\eta$ started to have non-zero input, the temperature kept on decreasing rapidly and it was close to 0 as $\eta$ reached maximum input values corresponding to the case studies ($\eta = 10$ for Figure 9a, $\eta = 6$ for Figure 9b). The lowest temperature values could also be linked with the velocity profiles obtained in Figure 8, which showed that in similar $\eta$ values, the velocity reached 0, thus almost demobilizing the fluid within the boundary layer. Since there was no velocity or marginal velocity recorded at $\eta = 10$ in Figure 8a and $\eta = 6$ in Figure 8b, this backs up the concept that temperature will be close to 0 or equal to 0 as the fluid remains static. In addition, the influential impact of the suction parameter $\xi$ largely reduced the temperature of the fluid, and the impact of $\varepsilon$ was quite marginal, indicating the dominance of $\xi$ (Figure 9b). While the curves seemed to overlap in Figure 9b, the magnified frame demonstrated the marginal differences at the narrowed axes.

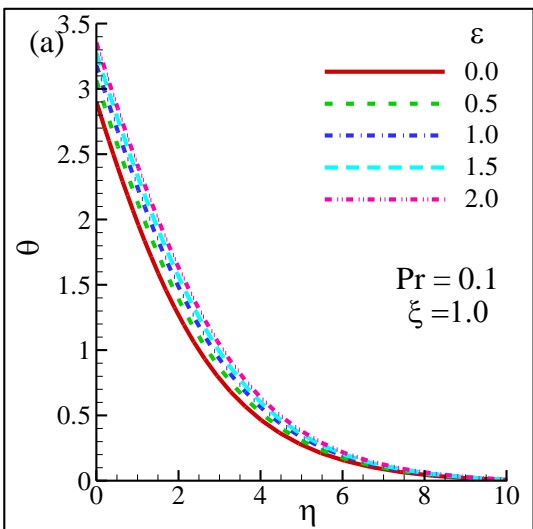 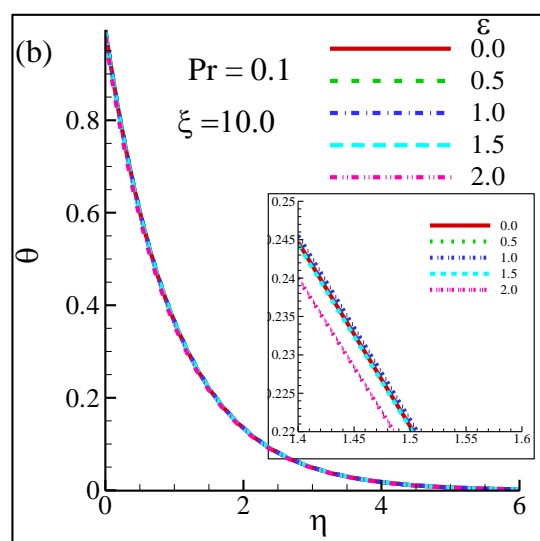

**Figure 9.** Temperature distribution for different viscosity-variation parameters $\varepsilon$ at (**a**) suction parameter $\xi = 1$ and (**b**) suction parameter $\xi = 10$ while $Pr = 0.1$.

### 3.7. Impact of $\varepsilon$, $\eta$, and $\xi$ Fluid Characteristics at Variable Pr

#### 3.7.1. Impact on Viscosity Distribution

In general, the $Pr$ number distinguishes different types of fluids. For example, $Pr = 0.01$ refers to sodium, $Pr$ between 0.02 to 0.03 refers to mercury. The purpose of different $Pr$ tests in this study is to explore the uniformity of the numerical solution.

Figure 10 shows the changes in the non-dimensional viscosity values as a function of $Pr$. The value of $\varepsilon$ was kept constant at 1, whereas two different suction parameters were appended in the numerical simulations. In addition, $\eta \in [0,10]$ was also considered. It was observed that the fluid corresponding to $Pr = 0.01$ exhibited the highest peak value when $\eta = 0$, indicating the absence of $\eta$ or the fact that the low viscosity-variation parameter increased the viscosity value. As $Pr$ got increased, the peak values got reduced as well. While the local maximum viscosity of the fluid of $Pr = 0.01$ was found to be 9.45 at $\eta = 0$, the fluid corresponding to $Pr = 0.1$ exhibited a local maximum viscosity of 4.2. However, as $\eta$ kept on increasing, the viscosity kept decreasing due to the dominance of $\eta$ on the viscosity values. However, these phenomena were obtained at a low suction parametric value $\xi = 1$, as presented in Figure 10a. As $\xi$ increased to 5, the peak values of each fluid at different $Pr$ exhibited a rapid increase in the local maxima. The fluid corresponding to $Pr = 0.01$ demonstrated a peak viscosity value of approximately 40, which was 9.45

when $\zeta = 1$ was considered. This 323.28% increase in the value could be attributed to the increase in $\xi$, which eventually increased the viscosity value in the absence of $\eta$. The rest of the fluids with further increased *Pr* also demonstrated similar characteristics, as depicted in Figure 10b.

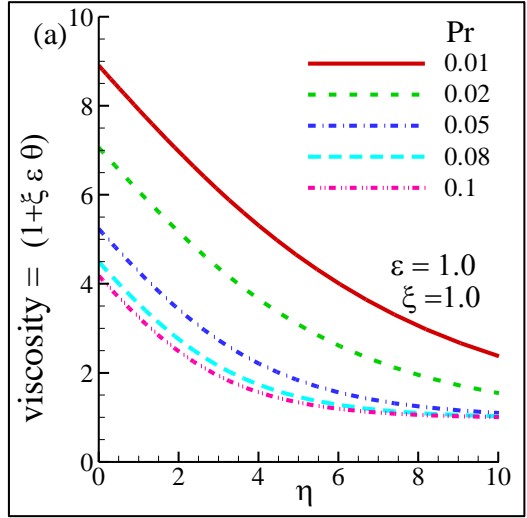 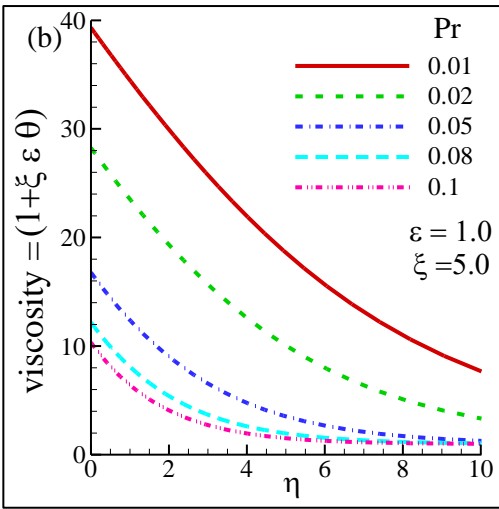

**Figure 10.** Viscosity distribution for the different Prandtl number (*Pr*): at (**a**) suction parameter $\zeta = 1$ and (**b**) suction parameter $\zeta = 5$ while $\varepsilon = 1.0$.

### 3.7.2. Changes in the Velocity Distribution

Velocity distribution corresponding to different *Pr* implies the influence of *Pr* on different types of fluids, as other parameters, such as $\eta$ and $\zeta$, are varied. The viscosity-variation parameter $\varepsilon$ was fixed at 1. In general, the patterns of velocity distribution had similarities with those obtained in Figure 11, which also indicates consistency in the current approach. As $\eta$ started to increase from 0, the velocity started to increase rapidly before hitting the peak, followed by a sharp decrease in a fluid motion. However, within the $\eta \in [0,10]$ boundary, only fluids corresponding to *Pr* = 0.01, 0.02 remained mobile, although the values of $f'$ were significantly lower at $\eta = 10$, as shown in Figure 11a. On the other hand, the rest of the fluids headed towards the static condition within the boundary layer at $\eta = 10$, thus establishing the preeminence of $\eta$ over the fluid mobility. A similar behavior was observed at an increased $\zeta = 5$ (Figure 8b), but the influence on $f'$ was not robust. As $\zeta$ increased to 5 from 1 in Figure 11b, the local maxima decreased. While at $\zeta = 1$, *Pr* = 0.02, the local maximum was recorded to be approximately 1.4, and the value was found to be around 0.85 when $\zeta = 5$ was considered. It should be mentioned here that while $\zeta$ attenuated the peak values of the velocity, it also influenced the local maxima of fluids with lower *Pr* values to occur at higher $\eta$. For example, fluid corresponding to *Pr* = 0.02 had a peak at $\eta \approx 2$ at $\zeta = 1$ (Figure 11a), whereas it was found to be at $\eta \approx 3$ when $\zeta = 5$ (Figure 11b).

### 3.7.3. Effect on Temperature Distribution

In the final part of sensitivity analyses, the effect of dimensionless parameters $\eta$ and $\zeta$ has been investigated on the temperature distribution as *Pr* varied from 0.01 to 0.1 under no viscosity variation, i.e., $\varepsilon$ was constant 1 throughout this analysis. In general, it was expected that $\eta$ would have more influence on the temperature distribution with varied *Pr* numbers, as fluids' temperature would likely to be more influenced by *Pr* values, as it would characterize the type of fluid, and $\eta$ would be able to dominate, since the viscosity-variation parameter was kept constant. The suction parameter $\zeta$ was expected to have a marginal influence on the temperature distribution of fluids with low *Pr* (in this case 0.01

and 0.02) and a pronounced effect on the rest. In any case, the influence of $\xi$ will not be significantly large on temperature distribution.

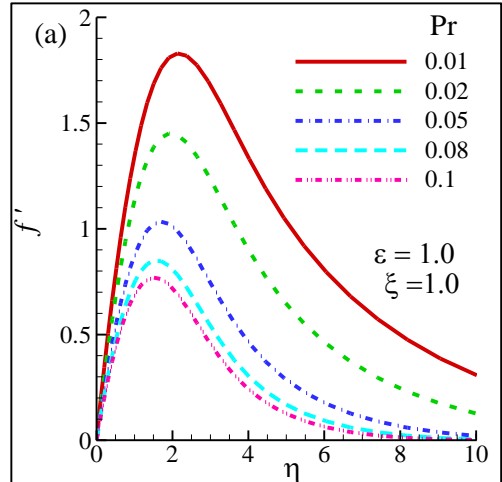 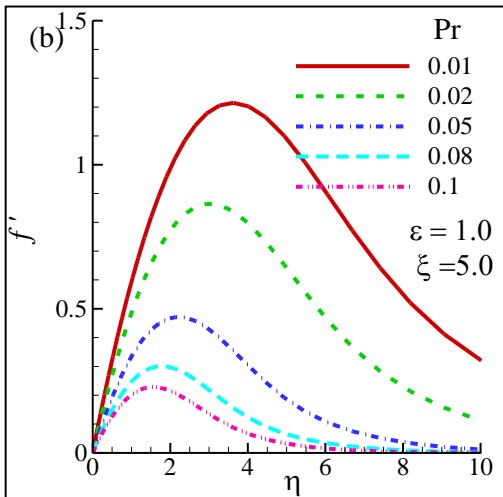

**Figure 11.** Velocity distribution for the different Prandtl number (*Pr*): at (**a**) suction parameter $\xi = 1$ and (**b**) suction parameter $\xi = 5$ while $\varepsilon = 1.0$.

Figure 12 supports the aforementioned claims. For instance, the fluid corresponding to *Pr* = 0.01 had the highest peak temperature value, regardless of the $\xi$ values. However, the peak was only visible at $\eta = 0$ in both cases ($\xi = 1$ or $\xi = 5$). As $\eta$ increased gradually, the temperature value $\theta$ kept decreasing, indicating a reduction in fluid temperature within the boundary layer. However, fluids with $Pr \geq 0.05$ exhibited a $\theta$ close to 0 as $\eta = 10$ was considered in the input. This behavior was also observed in both $\xi$ values. At *Pr* = 0.01, $\theta$ marginally decreased from 7.9 (Figure 12a) to 7.6 (Figure 12b) as $\xi$ increased from 1 to 5 when $\eta$ was non-existent in the computation. A similar attribute was noticed at *Pr* = 0.02 as well. However, at *Pr* = 0.1, the plummeting rate was significantly larger. While $\theta = 3.6$ was observed for *P r*= 0.1 under no $\eta$ condition (Figure 12a), the value was found to be decreased to 1.8 when $\xi = 5$ was considered. This decrease could be attributed to the thermal properties of the fluid. Increased *Pr* values imply the fluids which have low or decreasing thermal conductivity. Therefore, as the *Pr* increased, the fluid was not able to conduct heat significantly, which eventually decreases the thickness of the boundary layer.

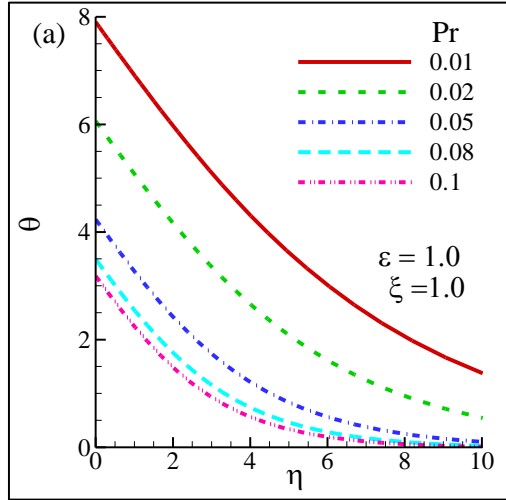 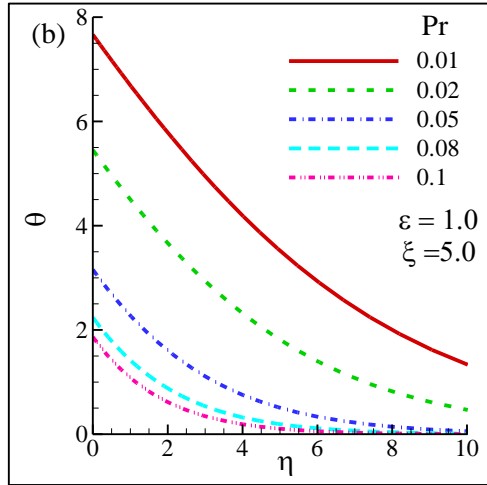

**Figure 12.** Temperature distribution for the different Prandtl number (*Pr*): at (**a**) suction parameter $\xi = 1$ and (**b**) suction parameter $\xi = 5$ while $\varepsilon = 1.0$.

## 4. Conclusions

The present study discusses the effect of viscosity-variation in the natural convective flow inside a permeable vertical cone with a uniform surface heat flux. The numerical analyses performed in this study will provide elaborate explanations on the novelty of considering temperature-dependent viscosity to aid in different environmental and agricultural applications targeting uniform heat flux of a porous medium such as soil. Soil contains different types of fluids and may be represented in different geometries to explore all possible scenarios that occurred by either natural or human-made phenomena. Different types of fluids have been assigned by varying *Pr* numbers. Later, further investigations have been conducted in different parametric studies, which could be subcategorized into two major categories: (i) fixed Prandtl number (*Pr*) and (ii) variable Prandtl number (*Pr*). The numerical model was developed by transforming the governing boundary layer equations into a dimensionless form. In addition, the local non-similarity equations, obtained as a result of a non-linear system of partial differential equations' reduction, were solved by three different solution techniques. The validations were conducted at both fixed *Pr* and variable *Pr* to assess the accuracy of the model. The model assessment was conducted by comparing the finite difference solution with the asymptotic solutions for large and small $\xi$ values, and good agreements were established. The summary of the present study is mentioned in the following:

- Increasing the suction parameter ($\xi$) leads to decreasing shear stress (local skin-friction coefficient) and increasing the rate of heat transfer (local Nusselt number). The increasing and decreasing characteristics could be attributed to the temperature difference of the fluid within the boundary layer, which requires balancing the physical difference.
- As the viscosity-variation parameter ($\varepsilon$) increases, the local skin-friction coefficient decreases concurrently due to the effect of viscosity. On the other hand, increasing the rate of heat transfer as a function of $\varepsilon$ does not remain consistent due to the effect of $\xi$. A small $\xi$ rate of heat transfer decreases as $\varepsilon$ increases, but the opposite behavior is observed with a large $\xi$, which indicates the superiority of the suction parameter over viscosity.
- If *Pr* increases, the local skin-friction coefficient decreases, and the rate of heat transfer increases rapidly due to the changes in the physical characteristics of the fluid and its viscosity.
- At $\varepsilon \neq 0$, as $\eta$ increases, viscosity decreases rapidly and heads towards 0 due to the dominance of $\eta$. However, at fixed $\varepsilon$ and variable *Pr*, the curves corresponding to viscosity values exhibit the lowest value at a much later stage, indicating the high viscous characteristics of fluids with low *Pr* number.
- At any value of $\varepsilon$, at $\eta \in [0,1]$, velocity increases the maximum at one point and then sharply plummets towards static condition at $\eta > 6$. However, a large suction parameter ($\xi = 10$) significantly lowers the peak values regardless of $\varepsilon$ or types of fluids.
- An increased value of $\varepsilon$ leads to the highest local maximum of temperature distribution in the absence of $\eta$ and at a fixed *Pr* number. However, at $\eta \neq 0$, the temperature starts to decrease gradually. The large suction parameter ($\xi = 10$) also suggestively lowers the peak values regardless of $\varepsilon$ or types of fluids. Meanwhile, at the variable *Pr* number, the local maxima of temperature distribution get marginally affected at a low *Pr* number (<0.05).

**Author Contributions:** Conceptualization, M.F.H. and M.M.M.; methodology, M.F.H. and M.M.M.; software, M.M.M.; validation, M.F.H. and M.M.M.; formal analysis, M.F.H.; investigation, M.F.H. and M.M.M.; resources, M.M.M., M.K., S.S.; data curation, M.M.M.; writing—original draft preparation, M.F.H. and M.M.M.; writing—review and editing, M.F.H., M.M.M., M.K. and S.S.; visualization, M.F.H. and M.M.M.; supervision, M.M.M.; project administration, M.F.H., M.M.M., M.K. and S.S.; funding acquisition, M.M.M. All authors have read and agreed to the published version of the manuscript.

**Funding:** The second author thanks the North South University (NSU) for the financial support as a Faculty Research Grant (CTRG-21-SEPS-12). The second author also acknowledges the Ministry of Science and Technology (MOST) of the government of Bangladesh for providing the financial support for buying a desktop computer (Grant No.:474-EAS).

**Institutional Review Board Statement:** Not applicable.

**Informed Consent Statement:** Not applicable.

**Data Availability Statement:** Data availability on request.

**Conflicts of Interest:** The authors declare no conflict of interest. The funders had no role in the design of the study; in the collection, analyses, or interpretation of data; in the writing of the manuscript, or in the decision to publish the results.

## Nomenclature

| | |
|---|---|
| $C$ | Specific heat |
| $C_f$ | Skin-friction |
| $f$ | Dimensionless stream function |
| $Gr$ | Grashof number |
| $k$ | Thermal conductivity |
| $Nu$ | Nusselt number |
| $Pr$ | Prandtl number |
| $q$ | Heat flux |
| $r$ | Radius of the cone |
| $u$ | Velocity component in the $x$-direction |
| $v$ | Velocity component in the $y$-direction |
| $x$ | Coordinate along a cone ray |
| $y$ | Coordinate normal to cone surface |
| $g$ | Acceleration due to gravity |
| $T$ | Temperature |
| $V$ | Transpiration velocity |

*Greek Symbols*

| | |
|---|---|
| $\beta$ | Thermal expansion coefficient |
| $\epsilon$ | Viscosity-variation parameter |
| $\gamma$ | Cone apex half-angle |
| $\Sigma$ | Summation |
| $\theta$ | Dimensionless temperature function |
| $\eta$ | Pseudo-similarity variable |
| $\xi$ | Dimensionless transpiration/suction parameter |
| $\rho$ | Fluid density |
| $\mu$ | Viscosity of the fluid |
| $\psi$ | Stream function |

*Subscripts*

| | |
|---|---|
| $fil$ | Film temperature |
| $i$ | Sequence of term |
| $p$ | Constant pressure or isobaric |
| $x$ | Differentiation with respect to $x$ |
| $w$ | Surface |
| $\infty$ | Ambient state |

*Superscript*

| | |
|---|---|
| ' | Differentiation with respect to $\widetilde{\eta}$ |
| $i$ | Number of iterations |

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
