# Peer review of "Natural Convection Flow over a Vertical Permeable Circular Cone with Uniform Surface Heat Flux in Temperature-Dependent Viscosity with Three-Fold Solutions within the Boundary Layer"

_computation, doi:10.3390/computation10040060_

Round 1

Reviewer 1 Report

I recommend accepting the manuscript after the authors do the major revisions included in the review report in the attachment file.

Referee's Report

Manuscript ID: computation-1665721

Journal: Computation (ISSN 2079-3197)

Title: Natural convection flow over a vertical permeable circular cone with uniform surface heat flux in temperature-dependent viscosity with three-fold solutions within the boundary layer

Recommendation: Major Revision

Reviewer's comments:

This paper discussed the effects of temperature dependent viscosity on natural convection flow from a vertical permeable circular cone with uniform heat flux. The governing boundary layer equations are transformed into a non-dimensional form and the resulting nonlinear system of partial differential equations are reduced to local non-similarity equations which are solved computationally by three distinct solution methodologies.

  1. The abstract can be written in a more interesting fashion with the significance of the study, the aim of the study, the research methodology, and the major conclusion of the study. Try to avoid the long sentences.
  2. Add some important numerical outcomes to the final abstract section.
  3. Nomenclature section is helpful in the manuscript.
  4. Why the authors chosen these three methods specifically, explain.
  5. The introduction section needs further improvements with the most recently related references to variable viscosity, and Keller box method.
  6. Some data and mathematical equations are mentioned in the manuscript need a help of references such as equations (1-4), (6), etc. Revise this.
  7. There have been several instances of punctuation mistakes. The manuscript should be reviewed because it contains some grammatical English errors. The manuscript requires proofreading.
  8. Clarify and discuss the motivation of this study, and the novelty and the significance of the results obtained here.
  9. Expand the discussion to highlight the relevance and interest of this work for its aimed scientific community.

The problem and results are sufficiently novel and interesting for possible publication in “Computation Journal”. I recommend accepting the manuscript with major revisions.

Author Response

I recommend accepting the manuscript after the authors do the major revisions included in the review report in the attachment file.

Recommendation: Major Revision

This paper discussed the effects of temperature dependent viscosity on natural convection flow from a vertical permeable circular cone with uniform heat flux. The governing boundary layer equations are transformed into a non-dimensional form and the resulting nonlinear system of partial differential equations are reduced to local non-similarity equations which are solved computationally by three distinct solution methodologies.

Reply: Thank you

==================================================================================

The abstract can be written in a more interesting fashion with the significance of the study, the aim of the study, the research methodology, and the major conclusion of the study. Try to avoid the long sentences.

Add some important numerical outcomes to the final abstract section.

Nomenclature section is helpful in the manuscript.

Why the authors chosen these three methods specifically, explain.

Reply: The abstract has been updated as advised. Changes are marked through “track changes”. The aim, research methodology, and major conclusion have been included. Long sentences have been re-written into shorter versions.

In the literature, most of the works consider constant viscosity of the fluid within the boundary layer and failed to prove the accuracy of the model in various ranges of Pr numbers. The current study considers both small (for example, 0.05) and large (for example, 0.7) Pr numbers. Therefore, it is essential to validate the current approach with different numerical parametric tests to gain more confidence on the accuracy and versatility of the approach. Therefore, more than one numerical solutions/validations should be con-ducted to gain more understanding on the accuracy. Among different numerical techniques, Keller-box method has been a popular and widely accepted numerical technique for nearly five decades [35], with more applications are being added into this scheme recently. For example, Kamran et al. [42] has considered the Keller-box approach to describe Jefferey-Hamel flow by considering different nondimensional parameters to obtain solutions. Reddy et al. [43] also obtained implicit finite difference results by the Keller-box technique to study the Joule heating and associated chemical reactions on magneto Cas-son nanofluid. Therefore, another evidence of expanding applications of Keller-box method was duly noted. On the other hand, perturbation and asymptotic solutions were also found to be highly efficient for small and large transpiration parameter (x), respectively for last few years [37]. Therefore, the validations through a combination of conventional and modern approaches should be adequate to prove the accuracy of the current approach and will provide a new dimension to the future study.

Please refer to L122-L166 for the updated information marked with “track changes”.

Nomenclature has been added before the introduction section as advised. It starts before the Introduction section for a better overview.

==================================================================================

The introduction section needs further improvements with the most recently related references to variable viscosity, and Keller box method.

Reply: The introduction section has been updated with some recent publications outlining the importance of variable viscosity, and Keller-box method.

Please refer to L85-L91, and L100-L106. The following recent research articles have been cited. In the reference section, they are numbered from [40] to [43].

  1. Khan, M.; Salahuddin, T.; Altanji, M. "A viscously dissipated Blasisus boundary layer flow with variable thermo-physical properties: An entropy generation study." Int. Comm. Heat Mass Transf. 131 (2022): 105873.
  2. Gladys, T.; Reddy, G.V.R. "Contributions of variable viscosity and thermal conductivity on the dynamics of non-Newtonian nanofluids flow past an accelerating vertical plate." Partial Diff. Eq. Appl. Math. (2022): 100264.
  3. Kamran, A.; Azhar, E.; Akmal, N.; Mehmood, Z.; Iqbal, Z. "Finite Difference Approach for Critical Value Analysis to Describe Jeffery–Hamel Flow Toward an Inclined Channel with Microrotations." Arabian J. Sci. Eng. (2022): 1-8.
  4. Reddy, Y.D.; Goud, B.S.; Chamkha, A.J.; Kumar, M.A. “Influence of radiation and viscous dissipation on MHD heat transfer Casson nanofluid flow along a nonlinear stretching surface with chemical reaction.” Heat Transfer (2022):1-17.

==================================================================================

Some data and mathematical equations are mentioned in the manuscript need a help of references such as equations (1-4), (6), etc. Revise this.

Reply: Done as advised. Changes are marked.

==================================================================================

There have been several instances of punctuation mistakes. The manuscript should be reviewed because it contains some grammatical English errors. The manuscript requires proofreading.

Reply: The whole manuscript has been proofread carefully and changes are marked for your reference. 

==================================================================================

Clarify and discuss the motivation of this study, and the novelty and the significance of the results obtained here.

Reply: The introduction has been updated with more explanations on the approach.

Please refer to L84-L98 for information on some industrial applications.

Refer to L139-L164 for the novelty of the current approach and why three-fold solutions were considered. 

The abstract has also been updated to provide more information on the approach.

Novelty of the approach has also been included in Conclusion L577-L583

==================================================================================

Expand the discussion to highlight the relevance and interest of this work for its aimed scientific community.

Reply: The applications in terms of porous material (soil) in environmental systems and agriculture have been specified in the revised Introduction section. Please refer to L84-L98

==================================================================================

The problem and results are sufficiently novel and interesting for possible publication in “Computation Journal”. I recommend accepting the manuscript with major revisions.

Reply: Thank you

==================================================================================

Reviewer 2 Report

This is a theoretical study investigating the natural convection over a vertical permeable circular cone immersed in a viscous fluid with temperature dependent viscosity. This study has possible applications in industry and agriculture.

A good documentation on the topic is made in introduction. The contribution of this work to the topic is that the model developed and validated here is also valid for fluids with low Pr numbers, in contrast with other studies in literature.

The validation was performed by three-fold numerical solutions and the results were very close.

The authors made a good discussion of the results, and results sustain the conclusions.

Just two observations:

-What is the meaning of w suffix and what is qw ? Please explain at the first appearance in the text;

-For audience, be more explicit about Eq. 1,2 and 3. What describes/represents each.

Author Response

This is a theoretical study investigating the natural convection over a vertical permeable circular cone immersed in a viscous fluid with temperature dependent viscosity. This study has possible applications in industry and agriculture.

A good documentation on the topic is made in introduction. The contribution of this work to the topic is that the model developed and validated here is also valid for fluids with low Pr numbers, in contrast with other studies in literature.

The validation was performed by three-fold numerical solutions and the results were very close.

The authors made a good discussion of the results, and results sustain the conclusions.

Reply: Thank you

==================================================================================

Just two observations:

-What is the meaning of w suffix and what is qw ? Please explain at the first appearance in the text

Reply: The suffix w refers to the surface. qw refers to surface heat flux. A complete nomenclature has been included prior to the Introduction section for clear understanding.

==================================================================================

-For audience, be more explicit about Eq. 1,2 and 3. What describes/represents each.

Reply: We have mentioned for equation (1) is the equation of continuity, momentum equation is (2) and energy equation is (3)

Round 2

Reviewer 1 Report

The authors have satisfactorily responded to all the questions and made the necessary changes to the manuscript. I have no further questions and suggest the acceptance of the revised manuscript.